# Potential of Traditional Chinese Medicine *Brucea javanica* in Cancer Treatment: A Review of Chemical Constituents, Pharmacology, and Clinical Applications

**DOI:** 10.3390/nu17203285

**Published:** 2025-10-20

**Authors:** Weiyin Xu, Hongmei Yang, Yanan Zhou, Rixin Guo, Jing Liu, Feng Wei, Yongqiang Lin

**Affiliations:** 1National Institutes for Food and Drug Control, Beijing 100050, China; xuweiyin1103@163.com (W.X.); zhouyanan@nifdc.org.cn (Y.Z.); guorixin@nifdc.org.cn (R.G.); linyongqiang@nifdc.org.cn (Y.L.); 2Public Experimental Center, Changchun University of Chinese Medicine, Changchun 130117, China; yanghm0327@sina.cn

**Keywords:** *Brucea javanica*, anti-cancer, chemical constituents, clinical application, molecular mechanism

## Abstract

*Brucea javanica *(BJ), a key representative of traditional Chinese herbal medicine, is derived from the dried mature fruit of *Brucea javanica* (L.) Merr., a plant in the Simaroubaceae family. Its pharmacological activity is largely attributed to diverse chemical constituents. To date, approximately 200 distinct chemical constituents have been isolated and identified, mainly comprising quassinoids, triterpenes, alkaloids, steroids, phenylpropanoids, and flavonoids. Contemporary pharmacological studies have demonstrated the significant activities of BJ in various areas, including anti-tumor, anti-inflammatory, and anti-parasitic effects. Notably, its oil form (*Brucea javanica* oil) has been extensively utilized in treating various cancer types. This review aims to systematically summarize the antitumor components, mechanisms of action, and clinical applications in cancer therapy, with the goal of providing theoretical support for further antitumor research and the development of new BJ-based drugs, highlighting its potential as an antitumor agent.

## 1. Introduction

Cancer, a highly destructive and heterogeneous disease, is primarily characterized by uncontrolled cell proliferation and the invasive behavior of abnormal cells [1]. Although approximately 5% to 10% of cancer cases are hereditary, the majority are closely linked to exogenous factors, such as environmental stimuli, dietary habits, lifestyle choices, and environmental pollution [2]. Cancer mortality rates have steadily increased, making it one of the leading global health threats [3]. Despite extensive research over the past decades into the treatment of malignant diseases, a definitive cure remains elusive. Current therapeutic approaches—including surgery, radiotherapy, chemotherapy, hormone therapy, immunotherapy, and targeted therapy—are limited by challenges such as systemic toxicity, severe side effects, drug resistance, high treatment costs, and low patient tolerance [4,5,6]. These factors significantly limit the acceptability and clinical applicability of these treatments. Therefore, the search for more efficient, selective, and novel therapeutic approaches to address the needs of various malignancies remains an urgent task.

Since antiquity, natural products and traditional Chinese herbal medicines have been extensively applied in cancer treatment and have played a crucial role in the discovery and development of anticancer agents [7,8]. Numerous studies have revealed that a vast array of natural products constitute a critical resource for anticancer agents, with over half of clinically used anticancer drugs derived from these substances [9,10,11]. For example, paclitaxel [12,13] and camptothecin [14], extracted from *Taxus wallichiana * var. *chinensis *(Pilger) Florin and *Camptotheca acuminata* Decne., respectively, have become pivotal drugs in anticancer therapy. Active components in Chinese herbal medicines—such as terpenoids, alkaloids, and flavonoids—exert anticancer effects through multiple mechanisms, including the modulation of signaling pathways, induction of apoptosis, and inhibition of angiogenesis. Clinical studies have demonstrated that the combined use of specific Chinese herbal medicines and chemotherapeutic agents can enhance therapeutic efficacy while mitigating adverse effects [15,16]. Furthermore, the structural diversity of natural products holds significant potential in cancer chemoprevention, providing novel insights for innovative drug design.

*Brucea javanica* (L.) Merr. (Simaroubaceae) is widely distributed in Southeast Asia and northern Australia. In China, it is primarily located in Fujian, Taiwan, Guangdong, Guangxi, Hainan, and Yunnan provinces [17]. Its dried mature fruits are commonly used as medicinal materials and hold significant importance in traditional Chinese medicine, exhibiting properties such as heat-clearing, dampness-drying, and detoxification. BJ demonstrates notable efficacy in treating dysentery, expelling parasites, and combating malaria, while also exhibiting antitumor activity against various cancers, including lung, rectal, esophageal, and colorectal cancers [17,18]. Additionally, it can be applied topically to treat skin lesions, such as warts and corns. Modern pharmacological studies have confirmed that BJ exerts significant inhibitory effects on multiple malignant tumors, particularly showing promising therapeutic outcomes in respiratory [19] and digestive system cancers [20,21]. Consequently, research on the active components of BJ and its anticancer properties has garnered increasing academic attention, with numerous researchers dedicated to elucidating its underlying molecular mechanisms. Since the 1970s, Chinese researchers have successfully developed *Brucea javanica* oil emulsion (BJOE) for intravenous administration as an anticancer agent. To date, *Brucea javanica* oil injection has been widely used in clinical practice, demonstrating significant therapeutic efficacy both as a standalone treatment [22] and in combination with radiotherapy [23] or chemotherapy agents [24]. *Brucea javanica* oil (BJO, also known as BJOE injection), a fatty oil extracted from the dried ripe fruits of *Brucea javanica* (L.) Merr., has become an important adjunctive therapy for patients with solid tumors during chemotherapy. This preparation enhances chemotherapy efficacy by increasing overall response rates, reducing postoperative adverse effects, and improving patients’ quality of life [25,26]. It is well established that chemotherapy agents often cause substantial harm to patients, with common adverse effects including leukopenia, hepatorenal impairment, and vomiting [27]. However, anticancer traditional Chinese medicine injections primarily serve as adjuvant therapies for tumor radiotherapy and chemotherapy, exhibiting functions such as toxicity reduction, symptom alleviation, and efficacy enhancement [28]. Furthermore, BJ-containing formulations, including capsules and oral preparations, have gained widespread clinical application as anticancer agents.

With the ongoing expansion of research in natural products, plant-derived bioactive compounds have become a focal point of investigation. Concurrently, the structural diversity of natural products has shown significant potential in cancer chemoprevention, providing novel insights for innovative drug design. In recent years, BJ has emerged as a research hotspot due to its remarkable antitumor bioactivity. Numerous studies have confirmed that BJ exhibits significant inhibitory effects against various malignant tumors, underscoring its high research value. Given the strong interest among readers for comprehensive information about BJ, this review systematically summarizes the antitumor-active chemical constituents isolated from BJ and their potential mechanisms of action. Furthermore, it elucidates the therapeutic roles of these compounds and provides an in-depth discussion of relevant clinical applications, aiming to offer theoretical support and references for further research, development, and clinical utilization of *Brucea javanica*.

## 2. Anticancer Chemical Ingredients

In recent years, the number of researchers dedicated to *Brucea javanica* (BJ) studies has increased significantly, with research primarily focused on bioactive molecules and their antitumor properties. With continuous advancements in analytical techniques, researchers have isolated over 200 natural compounds from BJ. These compounds can be broadly categorized into six major groups: quassinoids, alkaloids, triterpenes, steroids, phenylpropanoids, flavonoids, and others [21,29,30,31,32]. Notable examples include bruceine [33], brusatol [34], bruceantin [35], dehydrobrusatol [36], dehydrobruceantinol, dihydrobruceine, bruceantinol, bruceaketolic acid, yadanzioside [36,37], bruceoside, and bruceolide. Among these, quassinoids—the primary constituents of BJ—are widely recognized as the key bioactive components responsible for its anticancer activity. Notably, the most prominent quassinoids are classified as C20-quassinoids [38]. Their core structure consists of three hexagonal rings and a lactone ring. This class includes 110 compounds, with brusatol and bruceantin serving as representative examples exhibiting diverse pharmacological effects, including anti-inflammatory and tumor growth-inhibiting activities [39]. Research indicates that brusatol is primarily extracted from the seeds and fruits of *Brucea javanica*, where it constitutes approximately 0.3% of the fruit content [39,40]. It exhibits high concentrations and demonstrates significant antitumor activity against various cancers. The molecular structure of brusatol consists of an α, β-unsaturated cyclohexanone ring, two cyclohexane rings, a six-membered lactone ring, and a tetrahydrofuran ring [41]. Furthermore, studies on the pharmacophore moieties of the compound reveal that its activity primarily originates from the keto oxygen at the C-2 position, the enol oxygen at the C-11 position, and the epoxy methane bridge spanning C-8 to C-13 or C-8 to C-11 [42,43,44]. Figure 1 presents the structural schematic of anticancer active compounds in BJ.

## 3. Anticancer Effects

### 3.1. Anti-Lung Cancer

Lung cancer is the most frequently diagnosed malignancy in many countries and remains the foremost cause of cancer-related mortality worldwide [45,46]. Despite continuous progress in diagnostic and therapeutic strategies, patient survival rates remain poor. The discovery of multiple compounds in BJ that exhibit significant cytotoxic effects on lung cancer cells suggests that these components hold substantial potential for further investigation in lung cancer research. In the A549 lung cancer cell line, bruceanol series compounds demonstrated significant antitumor activity. Specifically, bruceanol C, E, D, and F had half-maximal inhibitory concentrations as low as 0.0064 μmol/L [47,48,49]. Additionally, bruceoside C, D, E, and F exhibit significant cytotoxicity in non-small cell lung cancer (NSCLC) cell lines [50,51]. Multiple studies have shown that bruceine B, 24-epipiscidinol A, yadanziolide B, quassilactone A and B, and bruceine D significantly inhibit the survival rate of H460 and A549 cells [29,49,52,53,54]. Cytotoxicity assays revealed that ethanol, petroleum ether, ethyl acetate, and n-butanol extracts of BJ exerted strong cytotoxic effects against the A549 cell line, with IC_50_ values spanning 0.02–17.47 μg/mL [55]. The water extract of BJ can induce apoptosis in non-small cell lung cancer (NSCLC) A549 cells, with an IC_50_ of 50 μg/mL [56]. Additionally, researchers observed that this extract exhibits specific inhibitory activity against the NSCLC cell line H1975, which carries the L858R/T790M epidermal growth factor receptor (EGFR) mutation [57]. Makong et al. [58] revealed that the methanol extract, dichloromethane, and ethyl acetate-soluble fractions of BJ root and bark, as well as compounds isolated from them (bruceacanthinones A, B et al.), exhibited significant cytotoxic effects on the A549 cell line. The IC_50_ values ranged from 50.0 ± 5.2 to 80.5 ± 1.8 μg/mL, with the cytotoxic effects of individual compounds being less potent compared to the crude extract. In A549 xenograft models, combined treatment with brusatol and cisplatin enhanced apoptosis, suppressed cell proliferation, and more effectively inhibited tumor growth than cisplatin alone [59].

### 3.2. Anti-Digestive System Cancer

Pancreatic cancer is a deadly disease characterized by aggressive tumor biology, with a higher prevalence in men [60]. Unfortunately, this disease is often asymptomatic, leading to diagnoses at advanced stages [61]. In vitro experiments on pancreatic cancer cells demonstrated that brusatol had IC_50_ values of 0.36 mmol/L for PANC-1 and 0.10 mmol/L for SW1990 cell lines [21], respectively. It also inhibited the proliferation of the PATU-8988 cell line and induced apoptosis [62]. The IC_50_ value of bruceine D for Hs68 cells exceeds 30 μmol/L [63], and it exhibits significant cytotoxic effects on Panc-1, SW1990, and Capan-1 cells, with antiproliferative effects comparable to those of the positive control drugs camptothecin and gemcitabine [64]. Bruceine A exhibited strong cytotoxic effects on human pancreatic cancer cell lines MIA PaCa-2, SW1990, PANC-1, and AsPC-1, inhibiting cancer cell proliferation in a time- and dose-dependent manner [65]. Furthermore, bruceine A caused a dose-dependent reduction in tumor growth in human pancreatic tumor-bearing mice, with its inhibitory effect at a dose of 0.5 mg/kg comparable to that of the positive control drug gemcitabine [65]. Moreover, Zhao et al. [21] demonstrated that bruceantinoside A, brusatol, yadanzioside A, yadanzioside C, bruceine D, bruceine H, and javanicoside G possess in vitro anti-pancreatic cancer activity, with brusatol showing the greatest potency.

Colorectal cancer is the third most commonly diagnosed malignancy worldwide and the second leading cause of cancer-related mortality, representing a major burden on global healthcare systems [66]. Tumor heterogeneity and clonal evolution during treatment contribute to frequent drug resistance issues [67]. However, natural products have garnered considerable attention due to their potent anticancer properties and relatively low adverse reaction rates. In 1993, Imamura et al. [47] demonstrated that bruceanol D, E, and F exhibited cytotoxicity against HCT-8 ileocecal cancer cells, with median effective dose (ED_50_) values ranging from 0.16 to 0.67 μg/mL. Subsequently, yadanziolide B, T, and bruceine B, D, E, and H exhibited cytotoxicity against HCT-8 cells, with IC_50_ values ranging from 1.3 to 6.7 μmol/L [29,68]. Numerous experimental studies in colorectal cancer treatment research have confirmed the sensitivity of the CT26 [69], HCT116 [70], PKO, SW480, and COLO205 cell lines to brusatol [71]. Notably, brusatol shows a marked cytotoxic effect on HCT116 cells when its concentration exceeds 15 nmol/L [70]. In CT26 cells, the IC_50_ value of this compound reaches 373 nmol/L [69]. Further studies indicate that cytotoxicity assessments of the SW480 cell line revealed IC_50_ values for brusatol, bruceine B, D, and yadanziolide A ranging from 0.1 to 28.5 μmol/L [43]. Based on these data, it can be concluded that brusatol exhibits the most potent cytotoxic effect on colorectal cancer cells among these monomeric compounds. Moreover, the IC_50_ values of the ethanol extract of BJ against HCT-116 and HT29 cells were 8.9 ± 1.32 μg/mL [72] and 48 ± 2.5 μg/mL [20], respectively. In in vivo experiments, brusatol administered at a dose of 2 mg/kg effectively inhibited the growth of xenograft and in situ tumors [69,71]. Notably, cisplatin and irinotecan, long-term treatment drugs for colorectal cancer, significantly enhance treatment efficacy when used in combination with brusatol or cisplatin [71,73].

Liver cancer is the sixth most frequently diagnosed malignancy worldwide and the fourth major cause of cancer-related mortality, with a five-year survival rate of only 18% [74]. Its high incidence and mortality rates contribute to its status as one of the most socially challenging global health issues [75]. Research on liver cancer has shown that brusatol, bruceine D, and B all exhibit significant cytotoxic effects. These studies are primarily conducted in vitro, with brusatol exhibiting the most significant cytotoxic effect in the SMMC7721 cell line, where the IC_50_ value is below 0.064 μmol/L. In contrast, bruceine B exhibits an IC_50_ value of 0.15 μmol/L for SMMC7721 hepatocellular carcinoma cells [49]. Furthermore, Ye et al. [76] reported that brusatol suppressed the proliferation of hepatocellular carcinoma cells in a dose-dependent fashion, with IC_50_ values of 0.69 μmol/L (Hep3B), 0.34 μmol/L (Huh7), 12.49 μmol/L (LM3), and 18.04 nmol/L (Bel-7404). Additionally, javanicolide H and E, bruceine B, E, H, and dehydrobrusatol exhibited cytotoxicity against HepG2 cells, with IC_50_ values ranging between 0.81 and 3.3 μmol/L [68]. Similarly, Yadanziolide T, B [29], together with bruceantinol [35], demonstrated marked growth-inhibitory activity in Bel-7402 and Bel-7404 cells, with IC_50_ values of 3.5–4.5 μmol/L and 10 μmol/L, respectively. In 2005, Lau et al. [56] reported that the aqueous extract of BJ triggered apoptosis in Hep3B cells, with an IC_50_ of approximately 50 μg/mL. A decade later, Chen et al. [77] further validated the cytotoxic activity of the BJ aqueous extract against the same cell line, observing an IC_50_ of 4 mg/mL. Notably, differences in dose–response effects of the BJ water extract may be attributed to inconsistencies in extraction methods.

Gastric cancer represents one of the most common malignancies worldwide, ranking as the fifth most frequently diagnosed cancer [78]. Because the majority of cases are identified at advanced stages, mortality remains substantial, positioning gastric cancer as the third primary cause of cancer-associated deaths. In the gastric cancer cell line BGC-823, yadanziolide T, yadanziolide B, javanicolide H, bruceine B, bruceine D, bruceine E, bruceine H, dehydrobrusatol, javanicolide E, and 24-epipiscidinol A all demonstrated significant cytotoxic effects. Among these compounds, javanicolide H showed the strongest cytotoxic effect, with an IC_50_ value of 0.52 μmol/L [29,52,68].

### 3.3. Anti-Reproductive System Cancer

Ovarian cancer is a highly aggressive malignancy, often diagnosed at advanced stages [79]. Despite initial surgery and chemotherapy, recurrence occurs in the majority of patients, underscoring the urgent need for novel therapeutic strategies. Globally, ovarian cancer—including malignancies of the ovary, fallopian tube, and peritoneum—accounts for an estimated 313,959 new cases and 207,252 deaths each year [80,81]. Studies exploring the application of BJ in reproductive system tumors have largely concentrated on breast, ovarian, and cervical cancers. In this field, preliminary investigations have examined the potential activity of compounds such as brusatol, bruceantinol, bruceine A, bruceine B, and brujavanol E against breast cancer. Furthermore, bruceine B, D, and H have been shown to exert notable cytotoxicity toward ovarian cancer cells. In addition, Bruceosides D, E, and F displayed marked and selective cytotoxic activity in ovarian cancer cell lines [51]. Subsequently, the cytotoxic effects of javanicolide E, bruceine B, D, E, H, and dehydrobrusatol were confirmed in SKOV3 cells, with half-maximal inhibitory concentration values ranging from 0.12 to 2.5 μmol/L [68].

Breast cancer is the most prevalent cancer among women globally. Characterized by distinct epidemiological features and marked heterogeneity, it continues to be a major cause of cancer-related mortality and poses a serious threat to women’s health [82]. Recent investigations have shown that compounds including brusatol, bruceantin, bruceines A, B, D, and E, brujavanol E, yadanziolide A, and yadanziosides G and B exert notable inhibitory effects on the MCF-7 human breast cancer cell line [35,37,49,83,84]. In the MDA-MB-231 cell line, brusatol, bruceantinol, bruceine A, and bruceantarin exhibited significant antitumor activity, with IC_50_ values ranging from 0.081 to 0.238 μmol/L [37]. Additionally, BJ hot water extracts induced apoptosis in the MDA-MB231 breast cancer cell line, with an IC_50_ around 50 μg/mL [56]. BJ ethanol extracts also demonstrated selective cytotoxicity and induced apoptosis at a concentration of 90 μg/mL [85]. A comprehensive analysis indicates that, compared to ethanol extracts, BJ water extracts demonstrate stronger inhibitory effects on the MDA-MB231 breast cancer cell line. In studies on cervical cancer, quassilactones A and B exhibited cytotoxic effects against HeLa cells, showing IC_50_ values of 78.95 ± 0.11 and 92.57 ± 0.13 μmol/L, respectively [86].

### 3.4. Anti-Leukemia

Leukemia represents a heterogeneous category of malignant blood disorders that arise from hematopoietic progenitor cells at different stages of hematopoietic maturation [87]. The etiology of leukemia involves genetic factors, environmental factors, and viral infections, typically triggered by monoclonal genetic or epigenetic abnormalities, but manifesting as a polyclonal disease. Bruceantin was first reported to exhibit anti-leukemia effects as early as 1973 [88]. Subsequently, the anti-leukemia effects of bruceantino, bruceantinoside A and B, brusatol, bruceanol A and B, and yadanzioside P were also documented [89,90,91,92,93]. In vitro experiments demonstrated that bruceoside C [50], yadanzioside G, N [94], bruceantinoside C [94], bruceanic acid D [95], and bruceanol C, D, E, and F [47,48] all exhibit potent cytotoxicity against P-388 lymphocytic leukemia cells, with ED_50_ values ranging from 0.16 to 7.49 μmol/L. Among these compounds, bruceanol D exhibited the highest safety profile. Additionally, reports indicate that javanicosides B, I, J, K, and L possess marked cytotoxic effects against P-388 murine leukemia cells, with IC_50_ values between 0.68 and 0.77 μmol/L [36,96]. Studies have shown that compounds including 17–18, (20R)-O-(3)-β-D-glucopyranosyl-(1→2)-α-L-arabinopyranosyl-pregn-5-en-3β, 20-diol, as well as brusatol, bruceines B, D, and E, yadanziolides A and C, yadanzigan, bruceoside A, and javanicolides B and S, display notable inhibitory effects on human promyelocytic leukemia (HL-60) cells [44,49,53]. Additionally, Liu [49] and colleagues demonstrated through in vitro experiments on the HL-60 cell line that brusatol and bruceine B exhibit significant inhibitory effects, with IC_50_ values of 0.06 and 0.27 μmol/L, respectively. Further studies indicate that brusatol and bruceantin have the potential to induce differentiation, inhibit proliferation, and exert differential cytotoxic effects across 11 leukemia cell lines. Experimental results indicate that brusatol and bruceantin exhibit lower cytotoxic responses in HL-60, K562, Kasumi-1, and Reh cell lines, but demonstrate extreme sensitivity in NB4, U937, BV173, SUPB13, RS4;11, Daudi, and DHL-6 cells, with significant cytotoxic effects [97]. In in vivo studies of P388 lymphocytic leukemia in mice, yadanzioside A, B, C, D, E, and G demonstrated significant antileukemic activity at a dose of 10 mg/kg, with an increased survival rate (ILS) ranging from 2.0% to 9.2% in experimental mice [98]. Yadanzioside P showed antileukemic effects at 5 and 10 mg/kg/day, elevating the ILS values of mice to 15.5% and 28.9%, respectively [93]. Similarly, Yadanzioside O exhibited activity at 2 and 4 mg/kg/day, with ILS increases of 37.1% and 47.2%, respectively [98]. As a monomer isolated from BJ, yadanzioside O provided the greatest survival benefit in animal studies.

### 3.5. Other Cancers

Beyond their pronounced antitumor effects in the tumor types discussed above, BJ compounds have been further investigated in the context of several other malignancies, such as oral cancer, nasopharyngeal carcinoma, glioma, renal cancer, medulloblastoma, and melanoma (Figure 2). Notably, as early as 1991, bruceantin, bruceolide, and bruceanic acid A were first identified by researchers [99]. Later investigations revealed that bruceoside C, bruceanol (D–G), and brujavanol (A, B, and E) displayed marked cytotoxic activity against human oral carcinoma cells, with IC_50_ values between 0.55 and 6.45 μmol/L [47,50,84,100]. Additionally, studies have demonstrated that BJ extract exhibits antiproliferative activity against KB and ORL-48 cells, with IC_50_ values of 24.37 ± 1.75 and 6.67 ± 1.15 μg/mL, respectively [101]. Based on the above findings, bruceanol G exhibits excellent cytotoxic activity in oral cancer. Bruceanol D, E, and F exhibited significant cytotoxic effects in TE-671 medulloblastoma cells, with ED_50_ values ranging from 0.14 to 0.22 μmol/L [47]. Further studies revealed that bruceanol demonstrates inhibitory activity against glioblastoma U87, U251, and MGG152 cells harboring gene mutations, particularly against IDH1-mutated U251 cells, with an IC_50_ value of approximately 20 nmol/L [102,103]. In investigations on head and neck squamous cell carcinoma (HNSCC) models, bruceanol markedly decreased cell viability, yielding IC_50_ values between 6 and 38 nmol/L across TU167, UMSCC47, UDSCC2, YD-10B, JMAR, FaDu, HN-9, and LN686 cell lines [104]. Bruceoside C, bruceanol D, E, and F [51] exhibited significant cytotoxic effects in melanoma cells (RPMI-7951), with ED_50_ below 0.15 μmol/L [47,50]. Additionally, bruceoside D, E, and F exhibit selective cytotoxicity against renal cancer cells, with log GI50 values ranging from −4.43 to −4.97 [51]. Among these, bruceanol D shows the lowest ED_50_, indicating a higher safety profile. Table 1 summarizes the IC_50_ values of various BJ compounds and extracts across different cancer cell lines, providing a clear comparison of their cytotoxic potencies.

## 4. Anticancer Mechanisms

### 4.1. Inducing Apoptosis

Apoptosis is a fundamental process in cancer pathophysiology and has a profound impact on the effectiveness of anticancer therapies. It can be triggered through two primary pathways: the intrinsic (mitochondrial) and extrinsic (death receptor) routes, both converging on the activation of the caspase family of cysteine proteases. Effector caspases, including caspase-3, -6, and -7, are responsible for executing the apoptotic program [106]. Multiple studies suggest that brusatol induces apoptosis primarily by decreasing mitochondrial membrane potential, increasing pro-apoptotic proteins such as Bax and Bak, and concurrently suppressing anti-apoptotic proteins including Bcl-2 and Bcl-xL, thereby facilitating cytochrome C release from mitochondria and activating the caspase cascade. This mitochondrial pathway induces apoptosis in various cell types, including pancreatic cancer (PANC-1, PATU-8988, Capan-2) [62,107], hepatocellular carcinoma (Bel-7404) [76], non-small cell lung cancer (PC9) [105], pituitary adenoma (GH3, MMQ) [108], nasopharyngeal carcinoma (CNE-1) [109], and head and neck squamous cell carcinoma (HNSCC, UD SCC2). In this pathway, caspase-9 activity and its downstream effector molecules (caspase-3, caspase-7, and poly (ADP-ribose) polymerase (PARP)) are activated. Concurrently, the expression levels of caspase-3 precursor (procaspase-3) and caspase-9 precursor (procaspase-9) decrease, while the levels of active caspase-3 (cleaved-caspase-3), caspase-8 (cleaved-caspase-8), and PARP (cleaved-PARP) significantly increase [73,76,109]. According to research, brusatol and bruceine D activate the JNK [110]/p38 MAPK [62] signaling pathways while suppressing the activation of Stat3/NF-κB, PI3K/Akt/mTOR [76], and PI3K/Akt/NF-κB [111] pathways, thereby inducing apoptosis in various tumor cells (Figure 3). These compounds reduce the protective effects of cancer cells by decreasing the activity of phosphorylated extracellular signal-regulated kinase (p-ERK), while simultaneously increasing the levels of phosphorylated p38 (p-P38) and phosphorylated c-Jun N-terminal kinase (p-JNK), thereby promoting the expression of pro-apoptotic proteins and inducing apoptosis [112]. In addition to regulating the aforementioned pathway proteins to induce apoptosis, bruceine D promotes apoptosis by suppressing the expression of microRNA-95 (miR-95), thereby upregulating the pro-apoptotic gene CUG triplet repeat RNA-binding protein 2 (CUGBP2) [113]. Furthermore, this compound downregulates P62 protein expression and upregulates microtubule-associated protein 1 light chain 3 (LC3) expression, thereby promoting apoptosis [76].

### 4.2. Inhibition of Cell Proliferation and Inducing Cell Cycle Arrest

In cancer treatment, inhibiting cell proliferation is one of the key strategies for anticancer therapy. Numerous anticancer drugs and treatments, including chemotherapy, targeted therapy, and immunotherapy, aim to control tumor growth and metastasis by suppressing tumor cell proliferation. However, the cell cycle, as the central process governing cell proliferation and division, is often dysregulated and is closely linked to cancer initiation, progression, and treatment [114]. The normal cell cycle consists of the G1, S, G2, and M phases, with regulation primarily depending on the precise control of cyclin proteins. Dysregulation of these regulatory mechanisms leads to uncontrolled cell cycle progression, resulting in unrestrained cell proliferation and ultimately tumor formation. As a result, cell cycle arrest has become a primary strategy for inhibiting tumor cell proliferation. Many anticancer drugs inhibit tumor growth by inducing cell cycle arrest. Research indicates that brusatol inhibits the Nrf2-Notch1 signaling pathway by downregulating Nrf2 and Notch1 protein expression, as well as their downstream target Hes1, thereby delaying cell proliferation [115]. The Cheng [116] research team demonstrated that bruceine D inhibits cell proliferation through the following five mechanisms: (1) promoting β-catenin protease cleavage, (2) inhibiting β-catenin protease and transcription factor-4 activity, (3) attenuating Wnt signaling pathway activity, (4) reducing jagged1 protein expression, and (5) inhibiting the Notch signaling pathway. Further research confirmed that bruceine D significantly reduces PI3K/Akt signaling pathway activity, inhibiting aerobic glycolysis [117]. This suppression targets hexokinase, phosphofructokinase, pyruvate kinase, and lactate dehydrogenase activity, thereby inhibiting cell proliferation. Crude BJEE and BJAE extracts induce cell cycle arrest by inhibiting the PI3K/Akt/mTOR signaling pathway [118] and elevating reactive oxygen species (ROS) levels in cancer cells [20], demonstrating their anticancer activity. Furthermore, BJAE induces EGFR expression to overcome drug resistance and promote cell cycle arrest at the sub-G1 phase [57]. Multiple experimental studies demonstrate that BJ extract significantly reduces the proportion of cells in the S phase while increasing the proportion of cells in the G0/G1 or sub-G1 phases. Reports indicate that brusatol, when combined with gemcitabine or 5-fluorouracil, induces G2/M phase cell cycle arrest in PANC-1 cells and enhances apoptotic effects [107]. Studies show that brusatol induces G1 phase arrest in melanoma cells [119]. In leukemia cell lines U937 and RS4:11, it induces G1 arrest and significantly increases the proportion of S phase cells, a phenomenon potentially associated with downregulation of c-Myc expression [97]. Brusatol induces breast cancer cells to arrest in sub-G0/G1 and G2/M phases by inhibiting Nrf2 [120]. In nasopharyngeal carcinoma studies, it induces G2/M phase cell cycle arrest by downregulating CyclinD1, Cdc2, and Cdc25c expression [109] (Figure 4).

### 4.3. Inhibition of Migration/Invasion

One of the hallmark features of many malignant tumors is their ability to migrate and invade [121]. Cancer cells detach from the primary tumor, invade the surrounding tissue, enter the circulatory system, and establish secondary tumor colonies in distant organs. The invasion–metastasis cascade is a multifaceted biological process, comprising several key stages: (1) local invasion of the basement membrane and cellular migration, (2) intravasation into blood vessels and/or the lymphatic system, (3) survival in the circulation, (4) arrest and extravasation at distant organs, and (5) colonization at the metastatic site [122,123]. Therefore, blocking cancer cell migration and invasion is considered a key strategy for cancer prevention and therapy. Studies indicate that BJ inhibits tumor cell invasion and metastasis through diverse mechanisms. First, BJ regulates epithelial–mesenchymal transition (EMT) in cancer cells, thereby diminishing their migratory and invasive abilities [124]. Specifically, Brucator inhibited the invasion and migration of liver tumors both in vivo and in vitro by modulating EMT, characterized by decreased N-cadherin and vimentin expression and increased E-cadherin expression [76]. Furthermore, Brucetin suppresses colorectal cancer cell metastasis by targeting the RhoA/ROCK1 signaling pathway and reversing EMT, as reflected in reduced expression of vimentin, N-cadherin, MMP2, and MMP9 proteins. In hepatocellular carcinoma, Brusatol reduced EMT markers such as fibronectin, vimentin, N-cadherin, Twist, and Snail, thereby suppressing STAT3-driven metastasis [125,126]. Second, angiogenesis is essential for cancer cell migration and metastasis. Vascular endothelial growth factor (VEGF), a downstream target of hypoxia-inducible factor 1 (HIF-1), is recognized as one of the most potent drivers of angiogenesis [127]. Brusatol induces the degradation of HIF-1α protein in HCT116 cells under hypoxic conditions, downregulates the expression of its downstream target VEGF, and consequently inhibits angiogenesis [70]. In gastric cancer studies, Brusatol reduces VEGF expression by inhibiting the Nrf2/HO-1 axis, thereby decreasing angiogenesis in cancer cells [128]. Third, multiple studies have reported that proteins such as transcription factor IIB-related factor 2 (BrF2) [129], insulin-like growth factor-binding protein 2 (IGFBP-2) [130], and CD151 [131] play regulatory roles in cancer cell growth, proliferation, migration, and invasion. The expression of proteins linked to metastasis is significantly diminished by brusatol in A549 cells, resulting in impaired migratory and invasive functions of cancer cells [132].

## 5. BJ Antitumor Clinical Preparations Research

Traditional Chinese medicine (TCM) has been employed in cancer therapy for centuries and has shown notable therapeutic outcomes [133,134]. The Chinese market has currently approved multiple BJO preparations for sale, including BJOE for injection, BJO soft capsules, BJO oral emulsion, and BJ pills, all of which have a wide range of clinical applications. These TCM formulations are primarily used as adjuvant therapies in cancer treatment, with their market sales showing a consistent upward trend. Considering factors such as market size, clinical application scope, and sales volume, the market performance of various formulations follows this hierarchy: BJOE injection > BJO soft capsules > BJO oral emulsion > other formulations. Current evidence on the safety and efficacy of BJOE and BJO in clinical settings is derived mainly from observational studies, underscoring the necessity for stronger evidence-based validation (Figure 5). Adverse reactions reported during clinical use include nausea, vomiting, liver damage, other digestive system injuries, pruritus, rash, skin lesions, and neurological impairments such as dizziness and headache [135,136]. These factors collectively limit the global promotion of BJO preparations. An analysis of the causes of adverse reactions associated with BJO preparations suggests potential toxicity from water-soluble saponins [137], as well as interactions between oleic acid in BJOE and solvents, which can lead to reduced formulation stability. Additionally, excipients like soy lecithin and glycerol can compromise the purity and stability of BJO and BJOE throughout manufacturing, storage, and utilization, thereby increasing the risk of adverse reactions. An additional concern is the excessive use of excipients, which complicates the detection of adulteration in formulations. Using electrospray ionization, researchers established an ultra-high-performance liquid chromatography–mass spectrometry (UHPLC-MS) method that enabled the identification of 69 components in BJO [138]. This technique not only effectively detects adulteration but also provides a solid theoretical foundation for establishing quality control standards for BJO.

TCM offers several advantages in multi-targeted therapy, including low drug resistance, verifiable efficacy, minimal side effects, and lower costs. Moreover, TCM effectively mitigates the toxic side effects of chemotherapy drugs. Current studies on BJO and BJOE have confirmed their cytotoxic activity against various tumor cell lines [139,140,141,142] (Figure 5). Researchers have conducted extensive investigations into the antitumor mechanisms of BJO and BJOE, which include the induction of apoptosis, disruption of the cell cycle, interference with cellular energy metabolism, inhibition of VEGF expression, suppression of tumor proliferation, reversal of drug resistance, enhancement of chemotherapy efficacy, reduction in drug toxicity and side effects, and delay of tumor progression [143,144,145,146]. Consequently, this drug not only inhibits and kills cancer cells but also enhances both cellular and humoral immunity without affecting normal cells [147].

Research indicates that BJOE significantly affects the survival rate, migration, and invasive capacity of esophageal cancer cells, while inhibiting their proliferation via the cyclin D1-CDK4/6 signaling pathway [148]. Additionally, BJOE exerts its antitumor effects by regulating the key factor MiR-8485, thereby modulating the LAMTOR3/mTOR/ATG13 signaling pathway to promote autophagy and apoptosis in ovarian cancer cells while inhibiting cell proliferation [146]. In investigating tumor autophagy, BJOE research has elucidated its mechanism of action in colorectal cancer cells. The study revealed that BJOE decreases LC3 protein expression, encompassing both LC3-I and LC3-II isoforms. Notably, in HCT116 colon cancer cells characterized by high basal autophagy, BJOE exerted dual actions by suppressing autophagy while simultaneously inducing apoptosis [141]. Another investigation demonstrated that BJOE reduced oncogene E6 expression in human cervical carcinoma SiHa cells in a dose-dependent fashion, thereby inducing apoptosis. In addition, BJOE markedly suppressed the growth of SiHa xenograft tumors, an effect likely mediated through modulation of the ERK/MAPK and NF-κB signaling cascades [149]. The study findings indicate that BJO significantly prolongs the survival of H22 ascites tumor-bearing mice. This effect is closely linked to the activation of miRNA-29b and p53-related apoptotic mechanisms, which involve downregulating Bcl-2 protein expression while upregulating Bax, Bad, and other protein levels, as well as the mitochondrial pathway, which reduces cytochrome C protein expression [150]. Su et al. [151]. identified a novel mechanism underlying BJO’s efficacy and safety profile. By modulating gut microbiota metabolism to alter the host’s amino acid composition, BJO activates mTOR and exhibits microbiota-dependent inhibition of MDA-MB-231 xenograft tumor growth in mice, with no toxicity observed in non-target organs. Researchers revealed the molecular mechanism by which BJO induces apoptosis in T24 bladder cancer cells through in vitro experiments [152]. The proposed mechanism appears to involve enhanced expression of caspase-3 and caspase-9 proteins, coupled with concurrent suppression of NF-κB and COX-2, ultimately leading to activation of the caspase-dependent apoptotic pathway. Additionally, modulation of p53 and cyclin D1 expression by BJO induces G0/G1 phase arrest in A549 and H446 cells, consequently suppressing the proliferation of these lung cancer cell lines. Subsequent investigations indicated that BJO may promote apoptosis in these two lung cancer cell lines by elevating ROS generation and initiating the mitochondrial caspase signaling cascade [142]. In leukemia, the mechanisms underlying the actions of BJO and BJOE have been partially clarified through experimental studies. These mechanisms include suppression of PI3K/Akt pathway activation and upregulation of downstream effectors such as p53, thereby triggering apoptotic processes [153]. Additionally, they downregulate c-FLIP (L/S), myeloid leukemia-1, Bcl-2, and XIAP expression, inducing apoptosis through mitochondrial and death receptor pathways [139].

However, evidence from multiple randomized controlled trials indicates that BJO is a promising adjunctive therapy for various cancers, particularly gastrointestinal tumors [154,155,156]. Meta-analyses further support this notion: BJOEI combined with conventional chemotherapy significantly increases the clinical total effective rate of gastric cancer patients, improves their physical condition, and effectively alleviates chemotherapy-related adverse reactions, such as gastrointestinal reactions, bone marrow suppression, and liver function impairment [26,157]. This suggests that BJOEI not only enhances the efficacy of chemotherapy but also improves patient tolerance and quality of life, thereby increasing overall treatment adherence. Moreover, other studies have evaluated the efficacy and safety of BJO as an adjuvant therapy for hepatocellular carcinoma. The results demonstrated that the addition of BJO to standard anticancer regimens may reduce the incidence and severity of chemotherapy- or surgery-related adverse events and improve patients’ clinical conditions and physiological parameters [158]. This evidence further highlights the favorable safety profile and therapeutic potential of BJO, offering clinicians a promising complementary option for the integrated management of liver cancer.

Overall, BJO appears to exert synergistic antitumor effects and toxicity-attenuating properties in the treatment of digestive system malignancies. Its mechanisms of action may involve the modulation of tumor cell apoptosis, inhibition of inflammatory responses, and enhancement of host immune function. Nevertheless, large-scale, multicenter, randomized controlled trials are still needed to further confirm its clinical efficacy and safety, as well as to elucidate its pharmacological mechanisms and pharmacokinetic characteristics. These efforts will provide a robust scientific foundation for the clinical translation and precise application of BJO and its derivatives in cancer therapy.

## 6. Conclusions and Outlook

Numerous studies indicate that traditional Chinese medicine has become a significant trend in the development of novel anticancer drugs. Brucea javanica, known for its heat-clearing and toxin-eliminating properties, has been utilized in traditional Chinese medicine for centuries. This herb not only demonstrates remarkable therapeutic effects against various tumors but also exhibits synergistic effects when combined with chemotherapy or radiotherapy. Consequently, it shows great promise for pharmaceutical development and clinical utilization, establishing itself as one of the most extensively applied herbal resources in traditional Chinese medicine. The present work seeks to deliver an extensive overview of the bioactive compounds, therapeutic potential, anticancer mechanisms, and clinical performance of BJ in tumor therapy. Through an in-depth analysis of existing literature and research, this study examines the bioactive components present in BJ, including alkaloids, berberidins, triterpenoids, steroids, phenylpropanoids, and flavonoids. These compounds demonstrate notable anticancer activity in experimental models, including both in vitro and in vivo systems. Moreover, this review investigates the possible anticancer mechanisms of BJ, covering diverse aspects such as induction of tumor cell apoptosis, modulation of the cell cycle, inhibition of DNA synthesis, suppression of proliferation, obstruction of angiogenesis, restriction of tumor growth, enhancement of immune responses, and reversal of multidrug resistance. BJ demonstrates significant antitumor activity by mediating multiple signaling pathways and regulating the expression of various proteins. Specifically, it inhibits proliferation, adhesion, invasion, and metastasis in various tumor cells by modulating signaling pathways such as JNK, HIF-1α, PI3K/Akt, Nrf2, Wnt, JAK2-STAT3, and EGFR. Moreover, through signaling pathways such as Nrf2, RhoA/ROCK, EGFR/PI3K/Akt, Bax, and PI3K/Akt/mTOR, BJ also suppresses tumor proliferation and metastasis. Simultaneously, it induces tumor cell apoptosis and inhibits proliferation through pathways such as PI3K/YAP1/TAZ, PI3K/Akt, EGFR, and NF-κB, demonstrating high safety. As a result, the clinical application of BJ is gradually expanding.

To date, the BJ preparations used in clinical practice remain predominantly BJOE and BJO, with no specific single effective component identified for clinical application. In clinical practice, BJ preparations are frequently administered alongside chemotherapeutic agents such as gemcitabine, 5-fluorouracil, carboplatin, gefitinib, and cisplatin. The diverse chemical composition of BJ preparations enables multifaceted approaches to combat tumor cells, making them applicable in the treatment of various cancers. Moreover, BJ preparations are increasingly used as adjuncts to radiotherapy and chemotherapy to reduce drug dosages, mitigate toxic side effects, and alleviate adverse reactions induced by these treatments. Future research should focus on the following areas: First, advanced methodologies—including artificial intelligence, organoid models, high-throughput screening, multi-omics approaches, and bioinformatics—should be employed to elucidate the anticancer pharmacological mechanisms of BJ, thereby constructing a comprehensive gene-signaling pathway framework to underpin clinical translation. Second, optimizing oral formulations by incorporating safer excipients with superior solubility can improve the physicochemical stability of both conventional and enteric-coated preparations while reducing the likelihood of adverse reactions. In clinical practice, there is an increasing demand for innovative dosage forms and diversified administration routes that can enhance patient compliance and therapeutic flexibility. Accordingly, future research should focus on developing advanced formulation strategies to improve the aqueous solubility and systemic bioavailability of BJ derivatives, which are currently constrained by poor solubility and extensive first-pass metabolism. Approaches such as nanoemulsion systems, solid lipid nanoparticles, and self-emulsifying drug delivery systems merit particular attention, as they can enhance intestinal absorption, prolong systemic circulation, and enable controlled or targeted drug release. Collectively, these formulation advancements are expected to improve in vivo efficacy, overcome existing pharmacokinetic limitations, and ultimately optimize the therapeutic potential of BJ-based anticancer agents. Third, BJ should undergo comprehensive preclinical and clinical studies in the future to confirm its anticancer mechanism, therapeutic effects, and safety profile. Particular emphasis should be placed on elucidating its pharmacokinetic and pharmacodynamic characteristics, as well as assessing its potential toxicity toward normal non-target cells and tissues to define a clear therapeutic window. Mechanistic studies are also required to clarify the pharmacological basis and structure–activity relationships of its active quassinoids and related compounds, thereby enabling rational optimization to enhance selectivity and minimize adverse effects. Furthermore, promising bioactive molecules should be isolated and subjected to systematic preclinical evaluation for efficacy and safety. In parallel, the development of advanced drug delivery systems—such as nanoparticle, liposomal, or targeted formulations—will be crucial for improving bioavailability, tumor specificity, and overall therapeutic performance. Fourth, current clinical studies primarily focus on short-term efficacy assessments, with limited long-term follow-up data. Future research should strengthen the establishment of long-term follow-up mechanisms to thoroughly assess patients’ long-term quality of life and tumor recurrence rates. Additionally, emphasis should be placed on developing new formulations of BJ and advancing their clinical translation processes to enhance their application value in cancer treatment. Current evidence suggests that BJ possesses considerable anticancer potential, highlighting its promising prospects for future applications in oncology. This review seeks to furnish researchers with a thorough reference for investigating the anticancer properties of BJ, while offering insights and theoretical guidance for advancing research and promoting the integration of traditional Chinese medicine into oncological practice. This will contribute to the development and integration of traditional Chinese medicine into the modern medical system.

## Figures and Tables

**Figure 1 nutrients-17-03285-f001:**
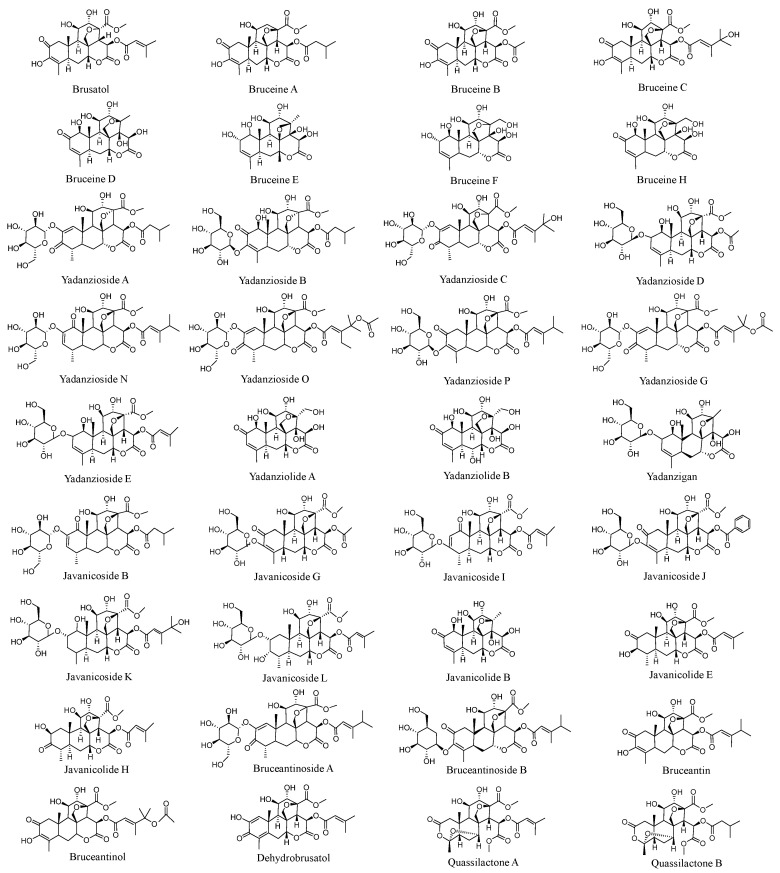
Chemical structures of anticancer active compounds in BJ.

**Figure 2 nutrients-17-03285-f002:**
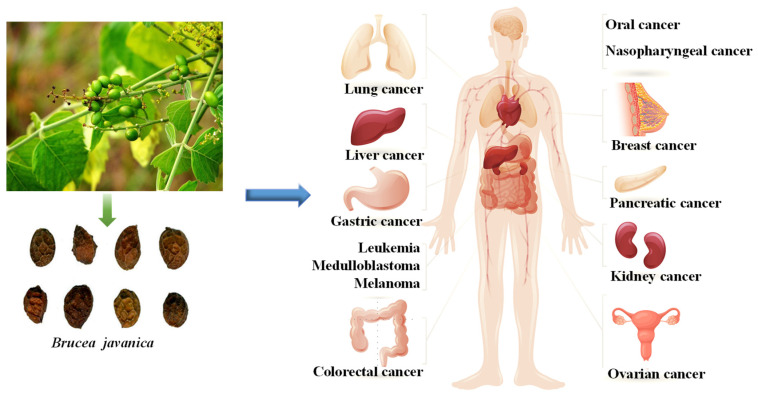
BJ shows significant antitumor effect against various cancers.

**Figure 3 nutrients-17-03285-f003:**
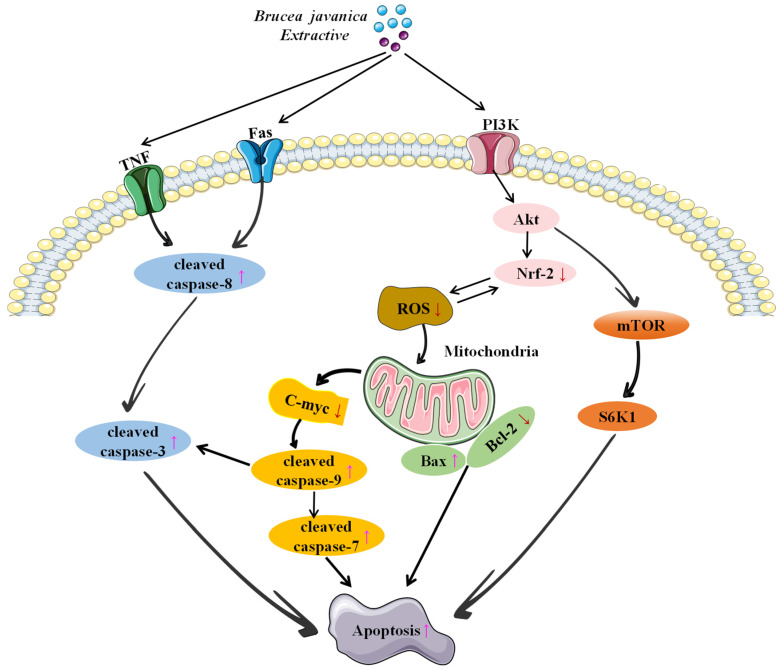
Molecular mechanism of BJ-induced cancer cell apoptosis.

**Figure 4 nutrients-17-03285-f004:**
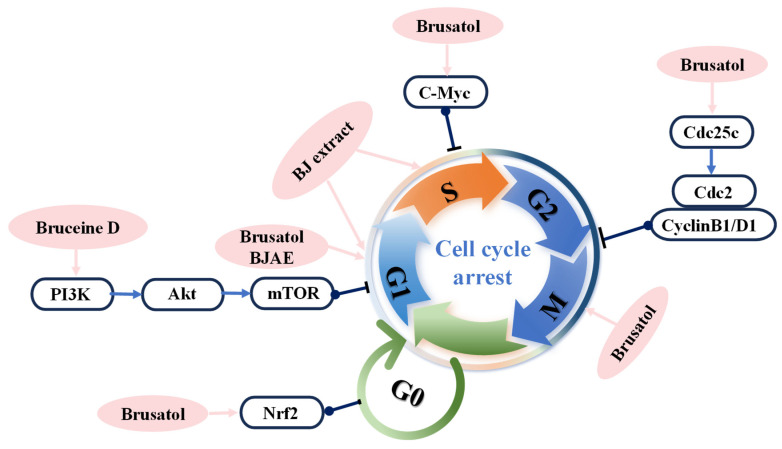
Molecular mechanism of BJ-induced cell cycle arrest.

**Figure 5 nutrients-17-03285-f005:**
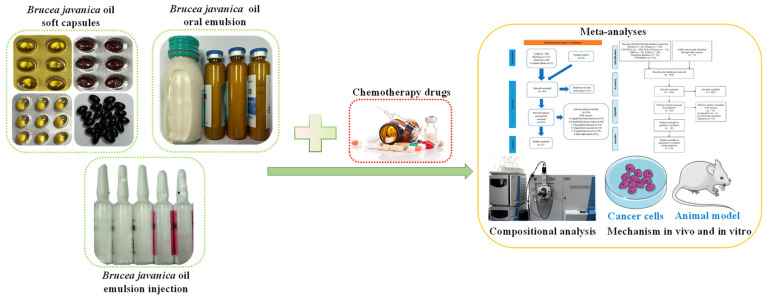
Clinical and experimental study of BJ-related preparations.

**Table 1 nutrients-17-03285-t001:** Cytotoxic activity (IC_50_ values) of BJ extracts and major bioactive compounds against various cancer cell lines.

Cancer Cell Line	Compound/Extract	IC50	Reference	Cancer Cell Line	Compound/Extract	IC50	Reference
HL-60	Brusatol	0.06 μmol/L	[49]	MCF-7	Brusatol	0.08 μmol/L	[49]
	Bruceine B	0.27 μmol/L			Bruceine B	0.54 μmol/L	
	Bruceine D	1.14 μmol/L			Bruceine D	6.75 μmol/L	
	Bruceine E	4.48 μmol/L			Bruceine E	17.77 μmol/L	
	Yadanziolide A	26.32 μmol/L			Yadanziolide A	14.61 μmol/L	
SMMC-7721	Brusatol	<0.064 μmol/L	[49]		BJ dichloromethane extract	55.1 μg/mL	[58]
	Bruceine B	0.15 μmol/L			BJ methanol extract	80.5 μg/mL	
	Bruceine D	0.88 μmol/L			BJ ethanol extract	15.12 μg/mL	[55]
	Bruceine E	4.27 μmol/L			BJ ethyl acetatel extract	3.28 μg/mL	
	Yadanziolide A	12.35 μmol/L			BJ petroleum ether extract	15.15 μg/mL	
A-549	Bruceine B	0.24 μmol/L	[49]		BJ n-butyl alcohol extract	30.92 μg/mL	
	Bruceine D	3.30 μmol/L		SW480	Brusatol	0.10 μmol/L	[49]
	Bruceine E	7.62 μmol/L			Bruceine B	0.30 μmol/L	
	Yadanziolide A	17.05 μmol/L			Bruceine D	7.78 μmol/L	
	BJ dichloromethane extract	50.0 μg/mL	[58]		Bruceine E	28.48 μmol/L	
	BJ methanol extract	75.2 μg/mL		PANC-1	Brusatol	0.36 mmol/L	[21]
	BJ water extract	50 μg/mL	[56]	SW1990	Brusatol	0.10 mmol/L	[21]
	BJ ethanol extract	8.79 μg/mL	[55]	Hs68	Bruceine D	>30 μmol/L	[63]
	BJ ethyl acetatel extract	0.02 μg/mL		HCT-8	Bruceine B	2 μmol/L	[68]
	BJ petroleum ether extract	9.14 μg/mL			Bruceine D	2 μmol/L	
	BJ n-butyl alcohol extract	17.47 μg/mL			Bruceine E	6.70 μmol/L	
Hep3B	Brusatol	0.69 μmol/L	[76]		Bruceine H	1.30 μmol/L	
	BJ water extract	50 μg/mL	[56]	HCT116	Brusatol	15 nmol/L	[69]
	BJ water extract	4 mg/mL	[77]		BJ ethanol extract	8.90 μg/mL	[72]
Huh7	Brusatol	0.34 μmol/L	[76]	CT26	Brusatol	373 nmol/L	[70]
LM3	Brusatol	12.49 μmol/L	[76]	HT29	BJ ethanol extract	48 μg/mL	[20]
Bel-7404	Brusatol	18.04 nmol/L	[76]	BGC-823	javanicolide H	0.52 μmol/L	[68]
HepG2	Bruceine B	0.81 μmol/L	[68]	SKOV3	Bruceine B	0.12 μmol/L	[68]
	Bruceine D	1.2 μmol/L			Bruceine D	0.76 μmol/L	
	Bruceine E	2.9 μmol/L			Bruceine E	2.2 μmol/L	
	Bruceine H	2.8 μmol/L			Bruceine H	0.33 μmol/L	
	javanicolide H	>10 μmol/L			javanicolide H	0.23 μmol/L	
	javanicolide E	>10 μmol/L			javanicolide E	1.49 μmol/L	
Bel-7402	Yadanziolide B	4.24 μmol/L	[29]	MDA-MB231	BJ water extract	50 μg/mL	[56]
Bel-7404	Bruceantinol	10 μmol/L	[35]		Brusatol	0.081 μmol/L	[37]
P-388	javanicoside B	5.6 μg/mL	[96]		Bruceantinol	0.088 μmol/L	
	javanicolides C	>100 μg/mL			Bruceine A	0.228 μmol/L	
	javanicolides D	18 μg/mL			Bruceantarin	0.238 μmol/L	
	javanicosides C	18 μg/mL		HeLa	quassilactones A	78.95 μmol/L	[86]
	javanicosides D	89 μg/mL			quassilactones B	92.57 μmol/L	
	javanicosides E	16 μg/mL		KB	BJ extract	24.37 μg/mL	[101]
	javanicosides F	50 μg/mL		ORL-48	BJ extract	6.67 μg/mL	[101]
H1650	Brusatol	24.27 ng/mL	[105]	U251	Bruceanol	20 nmol/L	[102]
PC9	Brusatol	18.40 ng/mL	[105]	HCC827	Brusatol	73.3 ng/mL	[105]

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
