# Peer review of "Potential of Traditional Chinese Medicine Brucea javanica in Cancer Treatment: A Review of Chemical Constituents, Pharmacology, and Clinical Applications"

_nutrients, 2025, doi:10.3390/nu17203285_

Round 1
Reviewer 1 Report
Comments and Suggestions for Authors
The manuscript provides a comprehensive review of Brucea javanica (BJ), a traditional Chinese medicinal plant, for its potential in cancer treatment. It summarizes the plant's chemical constituents and their specific anti-cancer effects on various tumour types. The review elucidates the molecular mechanisms of BJ, highlighting its ability to induce apoptosis, inhibit cell proliferation, and suppress migration. It also discusses the clinical applications of BJ preparations as an adjunct therapy. Finally, the manuscript outlines future research directions to optimize BJ formulations, elucidate its pharmacological basis, and advance its clinical translation.
The manuscript, which details the bioactive compounds of BJ and their pharmacological effects in cancer treatment, falls well within the scope of Nutrients, particularly under the subject areas of bioactive nutrients and nutraceuticals.
Overall, the manuscript is well-structured and an excellent review for anyone interested in the topic. However, there are some areas where the manuscript could be improved for clarity and impact.
- Currently, the ICâ‚…â‚€ data are scattered throughout the text, which makes it difficult for readers to compare the potency of different compounds or the sensitivity of various cancer cell lines. I would highly recommend that the authors create a comprehensive table to present the ICâ‚…â‚€ values for each compound or extract across the different cancer cell lines discussed. This will significantly enhance the manuscript's utility.
- The current rendering of Figure 5 is unreadable due to poor resolution. Even when magnified, the text remains blurred and indistinct, making it impossible to interpret the figure's content. The authors must replace this figure with a high-resolution version.
- The section titled '5. Clinical Studies' is misleading. It primarily discusses pre-clinical research, including in vitro studies on various cell lines and in vivo studies in animal models. This content does not belong in a section dedicated to human clinical trials. I recommend the authors reorganize the manuscript and move all in vitro and animal study data into the appropriate sections, such as '3. Anticancer Effects' and '4. Anticancer Mechanisms.' The '5. Clinical Studies' section should then be revised to focus exclusively on human clinical data. This could include details from clinical trials, observational studies, or case reports on the use of Brucea javanica preparations in cancer patients.
- The manuscript provides an excellent overview of the anti-cancer effects and mechanisms of Brucea javanica. However, their bioavailability, a significant pharmacological challenge, is not addressed. The primary active constituents, particularly the quassinoids, are often characterized by low water solubility and poor oral bioavailability, which can severely limit their clinical efficacy. I recommend that the authors include a section addressing this issue, discussing whether these extracts or their active ingredients have known bioavailability problems, if specific bioavailability or pharmacokinetic studies have been conducted for compounds like brusatol, and whether any formulation strategies are currently being explored to address this issue. I believe that this addition would provide a more complete and scientifically rigorous review, bridging the gap between the plant's therapeutic potential and its clinical application.
Author Response
Please see the attachment.

Responses to reviewers' comments (ID nutrients-3907395)
Dear editor and reviewers,
The authors appreciate both editor and reviewers for valuable comments and suggestions. All comments have been responded. Revisions have been made accordingly, which are labeled with red mark. We believe that the revisions have significantly improved the quality of the manuscript.
Best regards,
Weiyin Xu
Reviewer #1:
1. Currently, the ICâ‚…â‚€ data are scattered throughout the text, which makes it difficult for readers to compare the potency of different compounds or the sensitivity of various cancer cell lines. I would highly recommend that the authors create a comprehensive table to present the ICâ‚…â‚€ values for each compound or extract across the different cancer cell lines discussed. This will significantly enhance the manuscript's utility.
Response: Thank you for your valuable feedback on our manuscript. We completely agree with your suggestion that presenting the ICâ‚…â‚€ data in a table format would significantly enhance the readability and utility of the manuscript, making it easier for readers to compare the potency of different compounds and the sensitivity of various cancer cell lines. In response to your recommendation, we have added a comprehensive table to the manuscript, listing the ICâ‚…â‚€ values for each compound or extract across the different cancer cell lines. This table clearly presents the potency differences of each compound in various cell lines, enabling readers to make direct comparisons. We have also updated the relevant sections of the manuscript to include a description of the table. We believe this addition will greatly improve the manuscript's utility. Thank you again for your thoughtful review and suggestions. Should you have any further comments or recommendations, we would be happy to make additional improvements.
2. The section titled '5. Clinical Studies' is misleading. It primarily discusses pre-clinical research, including in vitro studies on various cell lines and in vivo studies in animal models. This content does not belong in a section dedicated to human clinical trials. I recommend the authors reorganize the manuscript and move all in vitro and animal study data into the appropriate sections, such as '3. Anticancer Effects' and '4. Anticancer Mechanisms.' The '5. Clinical Studies' section should then be revised to focus exclusively on human clinical data. This could include details from clinical trials, observational studies, or case reports on the use of Brucea javanica preparations in cancer patients.
Response: We sincerely thank the reviewer for the constructive and insightful comment. We have carefully reviewed and adopted the suggestion, revising the “5. Clinical Studies” section to address the inconsistency between its title and content. Following the reviewer’s recommendation, we have adjusted the title of this section to better reflect its actual scope. This section now aims to summarize and synthesize studies related to Brucea javanica (BJ) clinical anticancer preparations. In addition to removing preclinical content, we have incorporated human clinical data, including clinical trials and observational studies. We greatly appreciate the reviewer’s valuable feedback, as this revision has significantly improved the logical coherence and scientific rigor of our manuscript.
- The current rendering of Figure 5 is unreadable due to poor resolution. Even when magnified, the text remains blurred and indistinct, making it impossible to interpret the figure's content. The authors must replace this figure with a high-resolution version.
Response: Thank you for your valuable feedback regarding the quality of Figure 5. We fully understand the importance of high-quality figures for the clarity of our manuscript and acknowledge that the current resolution of Figure 5 is insufficient, making the figure difficult to interpret. We sincerely apologize for this issue. In response to your suggestion, we have replaced Figure 5 with a high-resolution version. The new figure ensures that all text, labels, and details are clearly visible, even when magnified, without any distortion. We have updated the figure to meet the high-quality standards necessary for clear presentation and better readability.
4. The manuscript provides an excellent overview of the anti-cancer effects and mechanisms of Brucea javanica. However, their bioavailability, a significant pharmacological challenge, is not addressed. The primary active constituents, particularly the quassinoids, are often characterized by low water solubility and poor oral bioavailability, which can severely limit their clinical efficacy. I recommend that the authors include a section addressing this issue, discussing whether these extracts or their active ingredients have known bioavailability problems, if specific bioavailability or pharmacokinetic studies have been conducted for compounds like brusatol, and whether any formulation strategies are currently being explored to address this issue. I believe that this addition would provide a more complete and scientifically rigorous review, bridging the gap between the plant's therapeutic potential and its clinical application.
Response: We are deeply grateful to the reviewers for their positive assessment of this study and their insightful suggestions. We fully concur with the reviewers' emphasis on the critical importance of bioavailability issues concerning the active components of Brucea javanica, particularly the quassinoids -type compounds. Indeed, low water solubility and poor oral bioavailability represent critical challenges that must be addressed for its clinical application. However, as the reviewer noted, our study primarily focused on the anticancer activity and mechanisms of action of B. javanica. To maintain the paper's structural coherence and thematic focus, we did not dedicate a separate chapter to bioavailability. Nevertheless, we have specifically addressed this issue in the revised manuscript's Conclusion and Outlook " section (Page 27-28, Lines 578–592) of the revised manuscript. This discussion outlines the pharmacokinetic and formulation challenges associated with BJ's major constituents and highlights future opportunities to enhance bioavailability through formulation optimization and novel delivery systems. We are grateful for the reviewer's valuable suggestion, which has further strengthened the paper's integrity and scientific rigor.
Reviewer 2 Report
Comments and Suggestions for Authors
The manuscript entitled “Potential of Traditional Chinese Medicine Brucea javanica in Cancer Treatment: A Review of Chemical Constituents, Pharmacology, and Clinical Applications” gives an extensive overview of the antitumor components, mechanisms of action, and clinical applications of Brucea javanica in cancer therapy, to provide theoretical support for further antitumor research and the potential development of new drugs. This is a very comprehensive review article dealing with both the mechanistic and cancer type-specific impact of Brucea extracts.
Please indicate the nature of this review paper (narrative, systematic, or integrative). How were the papers selected? What were the inclusion-exclusion criteria for the paper selection? There is no search strategy at all. What databases and keywords were used? Please include the Search strategy paragraph and address these issues.
Authors could discuss the possible toxicity of the plant extracts to normal non-target cells if used for the treatment. Although numerous plant extracts often show good results in studies, there are always open questions regarding their potential toxicity on normal non-target cells and tissues, making this kind of toxicity one of the biggest obstacles to the possibility of an actual prescription drug.
Minor remarks:
Figure 5. – The right panel figure has a very small font; please enlarge.
Author Response
Please see the attachment

Responses to reviewers' comments (ID nutrients-3907395)
Dear editor and reviewers,
The authors appreciate both editor and reviewers for valuable comments and suggestions. All comments have been responded. Revisions have been made accordingly, which are labeled with red mark. We believe that the revisions have significantly improved the quality of the manuscript.
Best regards,
Weiyin Xu
Reviewer #2:
1. Please indicate the nature of this review paper (narrative, systematic, or integrative). How were the papers selected? What were the inclusion-exclusion criteria for the paper selection? There is no search strategy at all. What databases and keywords were used? Please include the Search strategy paragraph and address these issues.
Response: We sincerely thank the reviewer for this valuable comment and for emphasizing the importance of transparency in literature selection and review methodology. The primary aim of this work is to provide a comprehensive overview and critical synthesis of current research progress on Brucea javanica and its anticancer effects, as well as to discuss potential pharmacological mechanisms and clinical perspectives. Therefore, this paper is designed as a narrative review, rather than a systematic or integrative review. Given the narrative nature of this article, we did not employ a predefined systematic search strategy (e.g., PRISMA workflow) or rigid inclusion–exclusion criteria. Instead, relevant literature was selected based on its scientific relevance, representativeness, and contribution to the topic, allowing for a broad and in-depth discussion of the current state of research. We have clarified this point in the revised manuscript by explicitly stating in the Introduction section (page 5-6) that the present study is a narrative review. We appreciate the reviewer’s thoughtful suggestion, and we believe this clarification helps readers better understand the scope and purpose of our manuscript.
2. Authors could discuss the possible toxicity of the plant extracts to normal non-target cells if used for the treatment. Although numerous plant extracts often show good results in studies, there are always open questions regarding their potential toxicity on normal non-target cells and tissues, making this kind of toxicity one of the biggest obstacles to the possibility of an actual prescription drug.
Response: We sincerely thank the reviewer for this insightful and constructive suggestion. We agree that the potential toxicity of Brucea javanica extracts to normal non-target cells is a crucial issue that must be carefully considered prior to clinical application. In the revised manuscript, this aspect has been discussed in the “Conclusion and Outlook” section (Page 28, Lines 592–604). Specifically, we emphasize that future investigations should undertake well-designed preclinical and large-scale clinical studies to confirm not only the anticancer mechanisms and therapeutic efficacy but also the safety profile of BJ and its bioactive constituents. Furthermore, we propose that in-depth studies should focus on elucidating the pharmacological basis, structure–activity relationships, and molecular mechanisms of its major antitumor compounds. Identifying and isolating the most promising active constituents, followed by systematic toxicological evaluation, will be essential for defining their therapeutic window and minimizing off-target effects. In parallel, the development of advanced formulation strategies and innovative drug delivery systems may help to improve bioavailability and safety, thereby accelerating the clinical translation of BJ-derived anticancer therapeutics. We appreciate the reviewer’s comment, which helped us to strengthen the discussion on this important topic and improve the scientific depth of the manuscript.
- Figure 5. – The right panel figure has a very small font; please enlarge.
Response: We would like to thank the reviewer for their careful feedback on the figure. In response to the reviewer’s suggestion, we have enlarged the font size in the right panel of Figure 5 to improve its readability. The updated figure now ensures better clarity while maintaining appropriate proportions and formatting. The corresponding changes have been made to Figure 5 in the manuscript.
Reviewer 3 Report
Comments and Suggestions for Authors
The manuscript presents a well-structured review with a clearly defined focus. The schemes are particularly commendable, being clear, instructive, and easy to follow, which enhances the readability and understanding of the subject matter.
Recommendations for improvement:
- References: It is recommended to include more recent and relevant references, particularly review articles published in reputable journals such as Phytomedicine and Frontiers in Pharmacology. This would further strengthen the scientific foundation of the manuscript.
- Schemes and Figures:
- Figure 2: The order of the presented cancer types could be improved. For example, grouping liver, gastric, and colorectal cancers together
- Figure 4: Please indicate more clearly the point at which bioactive compounds enter the cycle
- Figure 5: The meta-analysis is not readable due to low resolution. A higher-quality figure should be provided to ensure proper interpretation of the data.
The manuscript has the potential to make a valuable contribution to the field; however, certain technical and content-related improvements are necessary before.
Author Response
Please see the attachment.

Responses to reviewers' comments (ID nutrients-3907395)
Dear editor and reviewers,
The authors appreciate both editor and reviewers for valuable comments and suggestions. All comments have been responded. Revisions have been made accordingly, which are labeled with red mark. We believe that the revisions have significantly improved the quality of the manuscript.
Best regards,
Weiyin Xu
Reviewer #3:
1. Recommendations for improvement:
References: It is recommended to include more recent and relevant references, particularly review articles published in reputable journals such as Phytomedicine and Frontiers in Pharmacology. This would further strengthen the scientific foundation of the manuscript.
Schemes and Figures:
Figure 2: The order of the presented cancer types could be improved. For example, grouping liver, gastric, and colorectal cancers together
Figure 4: Please indicate more clearly the point at which bioactive compounds enter the cycle
Figure 5: The meta-analysis is not readable due to low resolution. A higher-quality figure should be provided to ensure proper interpretation of the data.
The manuscript has the potential to make a valuable contribution to the field; however, certain technical and content-related improvements are necessary before.
Response: We would like to thank the reviewer for their valuable and constructive comments on our manuscript. We have carefully considered and implemented the suggested revisions, which are outlined below:
References: In response to the reviewer’s suggestion, we have added more recent and relevant references, particularly review articles published in reputable journals such as Phytomedicine and Frontiers in Pharmacology. These references will further strengthen the scientific foundation of the manuscript and enhance its credibility.
Figures: Figure 2: We have reorganized the cancer types to group liver, gastric, and colorectal cancers together, improving the logical flow and clarity of the figure.
Figure 4: We have more clearly indicated the point at which bioactive compounds enter the cycle, ensuring a more accurate and informative depiction.
Figure 5: Regarding the low resolution of the meta-analysis figure, we have provided a higher-quality version to ensure proper interpretation of the data.
We believe these revisions will enhance the quality and scientific rigor of the manuscript. Once again, we appreciate the reviewer’s valuable suggestions.
Round 2
Reviewer 1 Report
Comments and Suggestions for Authors
I would like to thank the authors for their thorough response and for addressing all of the points raised in my initial review.
I am pleased to recommend that the manuscript is suitable for acceptance in its current form.